# 'When will this end? Will it end?' The impact of the March–June 2020 UK COVID-19 lockdown response on mental health: a longitudinal survey of mothers in the Born in Bradford study

Josie Dickerson ![ORCID],[1] Brian Kelly ![ORCID],[1] Bridget Lockyer,[1] Sally Bridges,[1] Christopher Cartwright,[1] Kathryn Willan,[1] Katy Shire ![ORCID],[1] Kirsty Crossley,[1] Maria Bryant ![ORCID],[2] Najma Siddiqi ![ORCID],[2,3] Trevor A Sheldon ![ORCID],[1,4] Deborah A Lawlor ![ORCID],[5,6] John Wright ![ORCID],[1] Rosemary RC McEachan ![ORCID],[1,7] Kate E Pickett ![ORCID],[2] on behalf of the Bradford Institute for Health Research COVID-19 Scientific Advisory Group

For numbered affiliations see end of article.

**Correspondence to**
Dr Josie Dickerson;
Josie.Dickerson@bthft.nhs.uk

## ABSTRACT

**Objectives** To explore clinically important increases in depression/anxiety from before to during the first UK COVID-19 lockdown and factors related to this change, with a particular focus on ethnic differences.

**Design** Pre-COVID-19 and lockdown surveys nested within two longitudinal Born in Bradford cohort studies.

**Participants** 1860 mothers with a child aged 0–5 or 9–13, 48% Pakistani heritage.

**Main outcome measures** ORs for a clinically important increase (5 points or more) in depression (eight item Patient Health Questionnaire (PHQ-8)) and anxiety (Generalised Anxiety Disorder Assessment (GAD-7)) in unadjusted regression analyses, repeated with exposures of interest separated by ethnicity to look for differences in magnitude of associations, and lived experience of mothers captured in open text questions.

**Results** The number of women reporting clinically important depression/anxiety increased from 11% to 20% (95% CI 10%–13%; 18%–22%) and from 10% to 16% (95% CI 8%–11%; 15%–18%), respectively. Increases in depression/anxiety were associated with loneliness (OR=8.37, 95% CI 5.70 to 12.27; 8.50, 95% CI 5.71 to 12.65, respectively); financial (6.23, 95% CI 3.96 to 9.80; 6.03, 95% CI 3.82 to 9.51), food (3.33, 95% CI 2.09 to 5.28; 3.46, 95% CI 2.15 to 5.58) and housing insecurity (3.29, 95% CI 2.36 to 4.58; 3.0, 95% CI 2.11 to 4.25); a lack of physical activity (3.13, 95% CI 2.15 to 4.56; 2.55, 95% CI 1.72 to 3.78); and a poor partner relationship (3.6, 95% CI 2.44 to 5.43; 5.1, 95% CI 3.37 to 7.62). The magnitude of associations between key exposures and worsening mental health varied between ethnic groups. Responses to open text questions illustrated a complex interplay of challenges contributing to mental ill health including: acute health anxieties; the mental load of managing multiple responsibilities; loss of social support and coping strategies; pressures of financial and employment insecurity; and being unable to switch off from the pandemic.

## Strengths and limitations of this study

► Three key longitudinal studies have highlighted that the COVID-19 pandemic and lockdowns have had a negative impact on mental health, particularly in younger adults, women and those from low socio-economic circumstances, but with participants of predominantly White European ethnicity.

► The Born in Bradford research programme offers a unique opportunity to investigate the impact of COVID-19 lockdown on mental health in a deprived and ethnically diverse population in whom mental ill health is often reported to be more prevalent.

► This is a longitudinal study containing linked data collected before the COVID-19 pandemic and during the March–June 2020 lockdown which has allowed us to explore change over that time period in a highly ethnically diverse population, the majority of whom live in the most deprived centiles in the UK.

► Respondents in this study were mothers of children aged 0–5 and/or 9–13 which may limit the wider generalisability, though our findings are broadly similar (in prevalence and associations) to those from another longitudinal study that included adult men and women.

► We are not aware of other studies that have explored longitudinal change in mental health from before to during the COVID-19 lockdown in a similar ethnically diverse and deprived population.

**Conclusions** Mental ill health has worsened for many during the COVID-19 lockdown, particularly in those who are lonely and economically insecure. The magnitude of associations between key exposures and worsening mental health varied between ethnic groups. Mental health problems may have longer term consequences for public health and interventions that address the potential causes are needed.

## INTRODUCTION

There is growing concern that the 'lockdown' measures to control the spread of the COVID-19 pandemic[1 2] have had unintended consequences including an increase in mental ill health. Several studies since the COVID-19 pandemic began have reported high levels of depression and anxiety[3–7] in the UK. However, some of these surveys are either cross-sectional or longitudinal *within* the lockdown period[6 7] with only three studies[3–5] comparing mental health in the pre-COVID-19 period to mental health during the COVID-19 lockdown. These studies consistently found worsening mental health in younger adults and those who were financially insecure. Other associations were reported in those who had a disability[3]; pre-existing mental and physical health conditions[4]; were living alone[4]; in parents of young children[5]; and in women.[4 5]

In these studies, most participants were of White European origin and the larger of the studies were in a relatively affluent population.[4] Mental ill health is often reported as being more prevalent in people from ethnic minorities and the socially and economically disadvantaged,[8 9] but no longitudinal research to date has investigated the impact of COVID-19 lockdown on mental health in these populations.[10]

We were able to explore these questions in depth by building on the Born in Bradford research programme which includes two longitudinal birth cohorts of ethnically diverse families living in the city of Bradford, many in deprived circumstances. These cohorts have recent in-depth information on the demographics, socioeconomic status and mental health of mothers before the COVID-19 pandemic,[11–13] as well as mental health during the March–June 2020 COVID-19 lockdown,[14] and so offer a unique opportunity to assess the impact of COVID-19 and lockdown longitudinally in a deprived and ethnically diverse population.

We used the data to:

► Describe the changes in the prevalence of depression and anxiety in mothers living in Bradford from before the COVID-19 pandemic to during the March–June 2020 COVID-19 lockdown.

► Identify variables associated with a clinically important increase in mental ill health in this population, to identify vulnerable groups that may need additional support in the recovery from the pandemic.

► Explore whether there is a difference in the magnitude of the associations of key exposures with an increase in clinically important mental ill health by ethnicity (Pakistani heritage compared with White British ethnicity).

► To explore mothers' lived experience during the March–June 2020 COVID-19 lockdown by assessing the frequency of worries and concerns relating to mental ill health obtained through free text responses to open questions.

## METHODS

### Study design

A longitudinal study using data collected at two time points, before and during the March–June 2020 COVID-19 lockdown, from mothers who participated in one of two prospective birth cohort studies in Bradford: Born in Bradford's Growing Up (BiBGU) study, with parents of children currently aged 9–13,[11 12] and Born in Bradford's Better Start (BiBBS), with parents of children currently aged 0–5.[13]

### Patient and public involvement

Born in Bradford is a 'people powered' research study; the local community was consulted to identify key research priorities during the March–June 2020 COVID-19 lockdown. This included consultation with key community groups, seldom-heard communities and local policy and decision makers to ensure that the focus of the research was relevant to local needs. The COVID-19 survey and recruitment approach were tested through our established research advisory groups. The findings of the study were also shared with these groups to enhance interpretation and ensure useful dissemination back to the community.[14]

### Consent

Participants had previously consented for their research data, and routinely collected health and education data, to be used for research. For the COVID-19 survey, verbal consent was taken for questionnaires completed over the phone and implied consent was assumed for all questionnaires completed via post or online.

### Data collection

Full details of the data collection of the March–June 2020 COVID-19 survey can be found elsewhere.[14 15] In summary:

Participants were recruited using a combination of emails, text and phone with a follow-up postal survey, and in their main language wherever possible.

#### Pre-COVID-19 data

Pre-COVID-19 data for BiBGU participants were derived from two sources: (a) participant ethnicity and age were collected during pregnancy (2007–2011)[11]; (b) recent follow-up data on mental health (collected between 24 June 2017 and 12 March 2020).[12] Pre-COVID-19 data for BiBBS participants were taken from data collected during pregnancy (6 January 2016 and 8 February 2020).[13] The median time since most recent pre-COVID-19 data collection was 15 months (range 1–35) for BiBGU and 29 months (range 2–52) for BiBBS.

#### March–June 2020 COVID-19 lockdown data collection

Mental health was measured using the eight item Patient Health Questionnaire (PHQ-8)[16] for depression and the Generalised Anxiety Disorder Assessment (GAD-7) for anxiety.[17] These are widely used measures of the severity of symptoms of depression and anxiety that have been

validated in the general population,[16 17] and in many ethnic minorities including UK residents of Pakistani heritage.[18] Information was also collected on household circumstances; family relationships; social support and loneliness; financial, employment, housing and food insecurity; and physical health. This contextual information was captured in self-reported questions administered in the COVID-19 lockdown survey; details of variables used are in table 1 and our protocol paper.[14] Information on the participants' lived experience during lockdown was captured using free text questions that asked 'What are your three biggest worries at the moment?' and 'Has anything been easier or more enjoyable in lockdown?'

## Data preparation

To describe the changes in prevalence of depression and anxiety from before the COVID-19 pandemic to during the COVID-19 lockdown, we derived a categorical variable for each PHQ-8 and GAD-7 score based on the standard clinical scoring classification: 0–4 no depression, 5–9 mild depression, 10–24 moderate-severe depression; 0–4 no anxiety, 5–9 mild anxiety, 10–21 moderate-severe anxiety,[16 17] we collapsed moderate, moderately severe and severe categories to indicate clinically important symptoms of depression and anxiety. To identify variables associated with an increase of mental ill health we used an increase in PHQ-8 and GAD-7 scores of 5 or more points as an indicator of clinically important change in symptoms. This cut-off was chosen following guidance from previous research and consultation with clinical colleagues.[19] An increase of 5 points or more would also always result in a change in the categorisation of symptoms (eg, from none to mild or mild to moderate/severe) while also capturing changes in severity within the moderate/severe categories.

Ethnicity was coded using Census 2011 categories and also as 'White British', 'Pakistani heritage' and 'Other', which included all other groups due to small numbers of a wide range of ethnicities. Given the heterogeneity of the 'Other' ethnic group, the decision was made to not use this group in these analyses, but to focus on comparing those of Pakistani heritage and White British ethnicity.

A number of categories within other explanatory variables were collapsed for analysis. These included: quality of relationship with partner: average to poor (comprising 'average', 'poor' and 'very poor'); loneliness: not lonely (comprising 'none or almost none of the time' and 'some of the time') and lonely (comprising 'Most of the time' and 'All or almost all of the time'); social support: easy to get support (comprising 'Very easy' and 'easy') and not easy to get support (comprising 'Very difficult', 'Difficult' and 'Possible'); food insecurity: secure (comprising 'never true' or 'sometimes true' that food did not last) and insecure (comprising 'Often true' that food did not last); housing security: secure (comprising 'Strongly Disagree', 'Disagree' and 'Neither disagree nor agree' that I worry about being evicted or having my home repossessed) and insecure (comprising 'Strongly Agree' or 'Agree').

**Table 1** Sample characteristics of mothers who responded to the BiB COVID-19 survey (n=1860)

| Financial security | | |
|---|---|---|
| Living comfortably | 354 | 20% (18%–22%) |
| Doing alright | 766 | 42% (40%–45%) |
| Just about getting by | 462 | 26% (24%–28%) |
| Finding it quite/very difficult | 223 | 12% (11%–14%) |
| Missing | 55 | |
| Change in financial status (compared with 3 months prior, as asked in lockdown survey) | | |
| Better off | 166 | 10% (8%–11%) |
| About the same | 989 | 58% (55%–60%) |
| Worse off | 562 | 33% (31%–35%) |
| Missing | 143 | |
| Index of Multiple Deprivation 2019 quintile | | |
| 1 (Most deprived) | 1211 | 65% (63%–67%) |
| 2 | 390 | 21% (19%–23%) |
| 3 | 139 | 7% (6%–9%) |
| 4 | 88 | 5% (4%–6%) |
| 5 (Least deprived) | 29 | 2% (1%–2%) |
| Missing | 3 | |
| Employment status of main earner | | |
| Employed—working from home | 393 | 22% (20%–24%) |
| Employed: going into work | 567 | 32% (30%–34%) |
| On furlough | 271 | 15% (14%–17%) |
| Self-employed: working | 147 | 8% (7%–10%) |
| Self-employed: not working | 200 | 11% (10%–13%) |
| Unemployed | 208 | 12% (10%–13%) |
| Missing | 74 | |
| Whether respondent is a key worker | | |
| No | 1259 | 68% (66%–71%) |
| Yes | 579 | 32% (29%–34%) |
| Missing | 22 | |
| Whether anyone in household is clinically vulnerable to COVID-19 | | |
| No | 1419 | 77% (75%–79%) |
| Yes | 426 | 23% (21%–25%) |
| Missing | 15 | |
| Whether anyone in household is currently/has previously been self-isolated | | |
| No | 1331 | 72% (70%–74%) |
| Yes | 508 | 28% (26%–30%) |
| Missing | 21 | |
| Children in household (n) | | |
| 1 | 387 | 21% (19%–23%) |
| 2 | 699 | 38% (35%–40%) |
| 3 | 451 | 24% (22%–26%) |
| 4 or more | 323 | 17% (16%–19%) |
| Total household size | | |
| 2/3 | 321 | 17% (16%–19%) |
| 4/5 | 943 | 51% (49%–53%) |
| 6 or more | 583 | 32% (29%–34%) |
| Missing | 13 | |
| Whether single parent | | |

**Table 1**  Continued

| | | |
|---|---|---|
| No | 1604 | 88% (86%–89%) |
| Yes | 222 | 12% (11%–14%) |
| Missing | 34 | |
| Quality of relationship with partner (for those married or in a relationship, n=1604) | | |
| Excellent | 783 | 51% (48%–53%) |
| Good | 598 | 39% (36%–41%) |
| Average | 129 | 8% (7%–10%) |
| Poor | 23 | 1% (1%–2%) |
| Very poor | 10 | 1% (0%–1%) |
| Missing | 61 | |
| Loneliness: how often felt lonely in the past week | | |
| None or almost none of the time | 998 | 57% (55%–60%) |
| Some of the time | 555 | 32% (30%–34%) |
| Most of the time | 131 | 8% (6%–9%) |
| All or almost all of the time | 56 | 3% (2%–4%) |
| Missing | 120 | |
| Social support: how easy to get help from friends/neighbours if needed | | |
| Very difficult | 101 | 6% (5%–7%) |
| Difficult | 146 | 8% (7%–9%) |
| Possible | 529 | 29% (27%–31%) |
| Easy | 529 | 29% (27%–31%) |
| Very easy | 513 | 28% (26%–30%) |
| Missing | 42 | |
| Food security: how often it is true that food does not last | | |
| Never | 1379 | 79% (77%–81%) |
| Sometimes | 279 | 16% (14%–18%) |
| Often | 91 | 5% (4%–6%) |
| Missing | 111 | |
| Housing security: whether worried about eviction/repossession | | |
| Strongly disagree or disagree | 1250 | 71% (68%–73%) |
| Neither | 328 | 19% (17%–20%) |
| Strongly agree/agree | 194 | 11% (10%–12%) |
| Missing | 88 | |
| Job security: whether they expect the main earner to still have a job in 12 months (for those currently employed, n=1652) | | |
| No | 77 | 5% (4%–6%) |
| Yes | 979 | 61% (59%–64%) |
| Don't know | 537 | 34% (31%–36%) |
| Missing | 59 | |
| Housing conditions: whether there is damp in the house | | |
| No | 1307 | 71% (69%–73%) |
| Yes | 534 | 29% (27%–31%) |
| Missing | 19 | |
| Frequency of physical activity currently | | |
| Never | 239 | 13% (12%–15%) |
| 1 or 2 days/week | 489 | 27% (25%–29%) |
| Most days | 516 | 28% (26%–30%) |
| Every day | 586 | 32% (30%–34%) |
| Missing | 30 | |
| Change in frequency of physical activity since lockdown | | |

Continued

**Table 1**  Continued

| | | |
|---|---|---|
| Less | 766 | 48% (46%–51%) |
| About the same | 408 | 26% (24%–28%) |
| More | 419 | 26% (24%–29%) |
| Missing | 267 | |

BiB, Born in Bradford.

Missing data on measures were small for most variables (table 1) and were not adjusted for in the analyses.

## Data analysis

To describe the changes in the prevalence of depression and anxiety from before to during the pandemic, we explored the changes in PHQ-8 and GAD-7 categories using descriptive statistics and presented these visually to elucidate the patterns of both positive and negative changes.

We used descriptive statistics to present the results of sample characteristics, including depression and anxiety scores at pre-COVID-19 and COVID-19 lockdown survey time points.

We then modelled the ORs associated with an increase in PHQ-8 and GAD-7 scores by 5 or more points using separate unadjusted logistic regression models for each exposure variable of interest. Exposure variables included in the analyses were identified as indicative of an increase in mental ill health in the lockdown survey findings reported previously.[15] Pre-COVID-19 PHQ-8 and GAD-7 scores were controlled for in each model.

In order to explore whether or not the magnitude of the association between exposure variables and a clinically important increase in symptoms of depression and anxiety differed between ethnic groups, we repeated the above analyses for Pakistani heritage and White British participants separately. This approach avoids the difficulties inherent in interpreting the ethnicity coefficient in regression models controlling for other variables.[20] All statistical analyses were carried out using Stata V.15.[21]

To explore mothers' lived experience during the March–June 2020 COVID-19 lockdown, the free text responses to open questions were reviewed and themes that related to mental health of the mothers were pulled out. Responses were coded using thematic analysis.[22] The first 100 responses were analysed by one researcher (BL), employing an inductive approach where coding and theme development were driven by the content of the responses. Two codebooks were developed, one for the questions on the three biggest worries and recent challenges during lockdown and another smaller codebook for the question on what had been made more enjoyable and easier during lockdown. Using Microsoft Excel, the remaining responses were then coded by three different researchers in order to test the strength and validity of the codebooks. Through frequent discussion between the researchers about this process, adjustments were made to the original codebooks so that they were reflective of the

total responses. The emergent themes relating to mental health were used to illuminate the findings from the quantitative analyses.

## RESULTS

### Study population

A total of 2144 (28%) of those invited participated in the COVID-19 survey between 10 April and 30 June 2020. Full details of the study population are described elsewhere and demonstrate that the population are broadly representative of the BiBGU and BiBBS cohorts.[15] Of these 2144 mothers, 1860 (87%) had complete surveys and linked data from pre-COVID-19 surveys and were used for this study. Of these participants, 1316 (71%) were in the BiBGU cohort (with a child aged 9–13) and 544 (29%) were in the BiBBS cohort (with a child aged 0–5) and had a mean age of 37.5 years (SD 6.8). Participants were ethnically diverse: 877 (48%) were of Pakistani heritage, 613 (34%) White British and 320 (18%) other ethnic groups; 1211 (65%) lived in the most deprived quintile of material deprivation in England. Table 1 describes the sample characteristics of the study population. Respondents were representative of the invited cohort on baseline (ie, prior to the COVID-19 pandemic) levels of depression and anxiety. Ethnicity was skewed with more White British and less Pakistani heritage respondents than in the invited sample, although the study sample still offers a large ethnically diverse population (see online supplemental table 1).

### Study sample change in prevalence of depression and anxiety (n=1860)

The prevalence of moderate/severe depression increased between the pre-COVID-19 and COVID-19 lockdown surveys (n=1860) from 11% (n=212, 95% CI 10% to 13%) to 20% (n=349, 95% CI 18% to 22%). The proportion of mothers reporting mild symptoms of depression remained similar while the proportion of those with no depressive symptoms decreased from 65% (n=1187, 95% CI 63% to 68%) to 56% (n=1001, 95% CI 54% to 59%) (table 2).

The proportion of mothers with moderate/severe anxiety increased from 10% (n=167, 95% CI 8% to 11%) to 16% (n=289, 95% CI 15% to 18%). The prevalence of mild anxiety also increased from 16% (n=270, 95% CI 14% to 18%) to 23% (n=408, 95% CI 21% to 25%), while the proportion of participants with no anxiety fell from 75% (n=1280, 95% CI 72% to 77%) to 61% (n=1075, 95% CI 58% to 63%) (table 2).

### Within-mothers change in depression (n=1760) and anxiety (n=1634)

A total of 1760 participants had both pre-COVID-19 and COVID-19 lockdown depression data and 1634 had pre-COVID-19 and COVID-19 lockdown anxiety data. Table 3 shows the change in depression and anxiety categorisations for these mothers across the two time points (see also online supplemental figures 1 and 2).

The majority of mothers stayed in the same depression and anxiety categories across the two time points, for example, 67% (n=759) who had no symptoms of depression in the pre-COVID-19 data continued to have no symptoms during lockdown; and 54% (n=109) of those who had moderate/severe symptoms of depression before the pandemic remained in this category in the lockdown survey. However, many mothers' mental health worsened, with 38% (n=230) who had no/mild symptoms of depression pre-COVID-19 reporting moderate/severe symptoms in the lockdown survey. A smaller number of mothers' mental health improved: of those mothers with moderate/severe symptoms of depression pre-COVID-19, 24% (n=48) subsequently reported no symptoms and 23% (n=46) reported mild depression in the COVID-19 survey. Similar patterns of change were seen with the anxiety categories.

### Factors associated with a clinically important increase in depression (n=1760) and anxiety (n=1634)

Three hundred and sixty-seven (21%) mothers reported a clinically important increase (5 or more points) in depressive symptoms, and 348 (21%) reported a clinically important increase (5 or more points) in anxiety symptoms.

**Table 2** The eight-item Patient Health Questionnaire (PHQ-8) depression scores and the Generalised Anxiety Disorder Assessment (GAD-7) scores in the pre-COVID-19 and COVID-19 surveys (n=1860)

| | Pre-COVID-19 survey | | COVID-19 survey | |
|---|---|---|---|---|
| | n | Percentage (95% CI) | n | Percentage (95% CI) |
| Depression category (PHQ-8) | | | | |
| None (PHQ-8 score 0–4) | 1187 | 65% (63% to 68%) | 1001 | 56% (54% to 59%) |
| Mild (PHQ-8 score 5–9) | 414 | 23% (21% to 25%) | 425 | 24% (22% to 26%) |
| Moderate to severe (PHQ-8 score 10–24) | 212 | 12% (10% to 13%) | 349 | 20% (18% to 22%) |
| Missing | 47 | | 85 | |
| Anxiety category (GAD-7) | | | | |
| None (GAD-7 score 0–4) | 1280 | 75% (72% to 77%) | 1075 | 61% (58% to 63%) |
| Mild (GAD-7 score 5–9) | 270 | 16% (14% to 18%) | 408 | 23% (21% to 25%) |
| Moderate to severe (GAD-7 score 10–21) | 167 | 10% (8% to 11%) | 289 | 16% (15% to 18%) |
| Missing | 143 | | 88 | |

**Table 3** Change in eight-item Patient Health Questionnaire (PHQ-8) and Generalised Anxiety Disorder Assessment (GAD-7) categories from pre-COVID-19 baseline to COVID-19 lockdown survey

| Pre-COVID-19 PHQ-8 | | | PHQ-8 at COVID-19 lockdown survey | | |
|---|---|---|---|---|---|
| Category | None | Mild | Moderate/severe | Missing | Total |
| None | 759 (67%) | **257 (23%)** | **118 (10%)** | 53 | 1187 (100%) |
| Mild | *171 (44%)* | 110 (28%) | **112 (28%)** | 21 | 414 (100%) |
| **Moderate/severe** | *48 (24%)* | *46 (23%)* | 109 (54%) | 9 | 212 (100%) |
| Missing | 23 | 12 | 10 | 2 | 47 (100%) |
| Total | 978 | 413 | 339 | 85 | 1860 (100%) |
| Pre-COVID-19 GAD-7 | | | GAD-7 at COVID-19 lockdown survey | | |
| Category | None | Mild | Moderate/severe | Missing | Total |
| None | 851 (70%) | **247 (20%)** | **118 (10%)** | 64 | 1280 (100%) |
| Mild | *95 (37%)* | 91 (36%) | **70 (27%)** | 14 | 270 (100%) |
| Moderate/severe | *47 (29%)* | *44 (27%)* | 71 (44%) | 5 | 167 (100%) |
| Missing | 82 | 26 | 30 | 5 | 143 (100%) |
| Total | 993 | 382 | 259 | 88 | 1860 (100%) |

Bold: category worse in lockdown compared with baseline. Italic: category improved in lockdown compared with baseline. Plain text: category stayed the same.

Table 4 presents the ORs for a clinically important increase in PHQ-8 and GAD-7 scores in relation to each exposure variable from the unadjusted logistic regression models. The estimates resulting from these models are imprecise, with wide CIs.

Financial, food and housing insecurity during lockdown was associated with a higher likelihood of a clinically important increase in both depression and anxiety symptoms: the odds were more than six times greater for women who were financially insecure (OR=6.23, 95% CI 3.96 to 9.8 for depression; OR=6.03, 95% CI 3.82 to 9.51 for anxiety) and over three times greater in mothers who were food insecure (OR=3.33, 95% CI 2.09 to 5.28 for depression; OR=3.46, 95% CI 2.15 to 5.58 for anxiety) or housing insecure (OR=3.29, 95% CI 2.36 to 4.58 for depression; OR=3.0, 95% CI 2.11 to 4.25 for anxiety).

Social circumstances were also associated with an increase in depression and anxiety: the odds of increased depression or anxiety were more than eight times greater in mothers reporting loneliness (OR=8.37, 95% CI 5.7 to 12.27 for depression; OR=8.5, 95% CI 5.71 to 12.65 for anxiety), and a lack of social support doubled the likelihood of an increase in depression and anxiety (OR=2.25, 95% CI 1.78 to 2.86; OR=2.13, 95% CI 1.67 to 2.73, respectively). An average/poor relationship increased the odds of experiencing symptoms of depression by 3.6 (95% CI 2.44 to 5.43) and of anxiety by 5.1 (95% CI 3.37 to 7.62). A larger household size reduced the OR for depressive symptoms (OR=0.73, 95% CI 0.53 to 0.99), but not for anxiety (OR=0.90, 95% CI 0.63 to 1.29).

A lack of physical activity during lockdown was associated with an increased OR for both depression and anxiety (OR=3.13, 95% CI 2.15 to 4.56; OR=2.55, 95% CI 1.72 to 3.78, respectively). Mothers who did the same amount of physical activity during lockdown as they had done pre-COVID-19 reduced the odds of an increase in depressive symptoms by 34% (OR=0.66, 95% CI 0.45 to 0.97).

There was no clear association between ethnicity (Pakistani heritage compared with White British) and an increase in depression (OR=0.88, 95% CI 0.68 to 1.15) or anxiety (OR=0.94, 95% CI 0.72 to 1.23). When the unadjusted regression analysis was repeated separately for White British and Pakistani heritage mothers, interesting differences in the magnitude of the association between exposure variables and a clinically important increase in symptoms of depression and anxiety were found, although CIs are wide (tables 5 and 6). The odds of an increase in depression and anxiety were greater for Pakistani heritage women who reported loneliness (OR=11.22, 95% CI 6.45 to 19.53 for depression; OR=11.27, 95% CI 6.16 to 20.56 for anxiety) compared with White British mothers (OR=7.17, 95% CI 3.66 to 14.01; OR=5.90, 95% CI 3.03 to 11.47, respectively). There was also a greater magnitude of association for increased depression, but not anxiety, for Pakistani heritage women who reported an average/poor relationship with their partner (OR=4.91, 95% CI 2.78 to 8.67 for depression; OR=4.99, 95% CI 2.72 to 9.14 for anxiety), compared with White British mothers (OR=2.61, 95% CI 1.31 to 5.20; OR=4.39, 95% CI 2.26 to 8.53, respectively).

In contrast, the magnitude of the odds of an increase in depression and anxiety was greater for White British women who reported financial insecurity (OR=12.14, 95% CI 5.15 to 28.60 for depression; OR=7.69, 95% CI 3.43 to 17.26 for anxiety) or never doing physical activity (OR=5.54, 95% CI 2.69 to 11.43; OR=3.90, 95% CI 1.82 to 8.36, respectively) compared with Pakistani heritage women (financial insecurity: OR=5.79, 95% CI 2.90 to 11.56 for depression; OR=4.35, 95% CI 2.27 to 8.30 for anxiety); (physical activity: OR=3.34, 95% CI 1.88 to 5.92; OR=2.50, 95% CI 1.40 to 4.45, respectively).

**Table 4** ORs (with 95% CI) from unadjusted logistic regression models for an increase of 5 points or more on eight-item Patient Health Questionnaire (PHQ-8) and Generalised Anxiety Disorder Assessment (GAD-7) between pre-COVID-19 and COVID-19 lockdown surveys

| Exposure | PHQ-8 increase ≥5 | | | GAD-7 increase ≥5 | | |
|---|---|---|---|---|---|---|
| | OR | Low 95% CI | High 95% CI | OR | Low 95% CI | High 95% CI |
| **Ethnicity (Reference: White British)** | | | | | | |
| Pakistani heritage | 0.88 | 0.68 | 1.15 | 0.94 | 0.72 | 1.23 |
| **Age group (Reference: Under 30 years)** | | | | | | |
| 30–34 | 1.01 | 0.66 | 1.54 | 0.90 | 0.57 | 1.42 |
| 35–39 | 0.89 | 0.59 | 1.35 | 0.99 | 0.64 | 1.53 |
| 40–44 | 1.04 | 0.69 | 1.58 | 0.88 | 0.57 | 1.38 |
| 45+ | 0.92 | 0.59 | 1.45 | 1.02 | 0.64 | 1.63 |
| **Financial security at pre-COVID-19 survey (Reference: Living comfortably)** | | | | | | |
| Doing alright | 1.07 | 0.81 | 1.42 | 1.19 | 0.89 | 1.59 |
| Just about getting by | 1.72 | 1.23 | 2.40 | 1.66 | 1.17 | 2.35 |
| Finding it quite/very difficult | 1.70 | 1.03 | 2.80 | 2.39 | 1.44 | 3.98 |
| **Financial security at lockdown survey (Reference: Living comfortably)** | | | | | | |
| Doing alright | 1.74 | 1.18 | 2.56 | 1.47 | 1.00 | 2.17 |
| Just about getting by | 3.52 | 2.36 | 5.26 | 3.20 | 2.15 | 4.76 |
| Finding it quite/very difficult | 6.23 | 3.96 | 9.80 | 6.03 | 3.82 | 9.51 |
| **Change in financial status compared with 3 months ago (Reference: Better off)** | | | | | | |
| About the same | 0.65 | 0.43 | 0.98 | 0.88 | 0.55 | 1.40 |
| Worse off | 1.36 | 0.89 | 2.01 | 1.99 | 1.25 | 3.19 |
| **Index of Multiple Deprivation 2019 quintile (Reference: 1—Most deprived fifth)** | | | | | | |
| 2 | 0.90 | 0.67 | 1.21 | 1.22 | 0.92 | 1.63 |
| 3 | 0.91 | 0.58 | 1.41 | 0.87 | 0.55 | 1.39 |
| 4 | 0.76 | 0.43 | 1.34 | 0.98 | 0.57 | 1.68 |
| 5 (Least deprived fifth) | 0.85 | 0.34 | 2.13 | 0.99 | 0.39 | 2.48 |
| Per one-fifth increase in lower deprivation | 1.07 | 0.94 | 1.25 | 1.00 | 0.88 | 1.13 |
| **Employment status of main earner (Reference: Employed and working from home)** | | | | | | |
| Employed: going into work | 1.18 | 0.84 | 1.65 | 1.20 | 0.85 | 1.69 |
| On furlough | 1.19 | 0.79 | 1.79 | 1.29 | 0.85 | 1.94 |
| Self-employed: working | 0.90 | 0.53 | 1.51 | 0.92 | 0.55 | 1.57 |
| Self-employed: not working | 1.49 | 0.97 | 2.27 | 1.31 | 0.84 | 2.04 |
| Unemployed | 2.03 | 1.34 | 3.09 | 2.23 | 1.44 | 3.44 |
| **Whether respondent is a key worker (Reference: No)** | | | | | | |

**Table 4** Continued

| Exposure | PHQ-8 increase ≥5 | | | GAD-7 increase ≥5 | | |
|---|---|---|---|---|---|---|
| | OR | Low 95% CI | High 95% CI | OR | Low 95% CI | High 95% CI |
| Yes | 0.99 | 0.77 | 1.27 | 1.05 | 0.82 | 1.35 |
| **Whether anyone in household is clinically vulnerable to COVID-19 (Reference: No)** | | | | | | |
| Yes | 1.68 | 1.29 | 2.19 | 1.53 | 1.17 | 2.01 |
| **Whether anyone in household is currently/has previously been self-isolated (Reference: No)** | | | | | | |
| Yes | 1.59 | 1.24 | 2.04 | 1.48 | 1.14 | 1.92 |
| **Number of children in household (Reference: 1)** | | | | | | |
| 2 | 1.20 | 0.87 | 1.67 | 1.05 | 0.76 | 1.45 |
| 3 | 1.39 | 0.98 | 1.98 | 1.20 | 0.84 | 1.71 |
| 4 or more | 1.12 | 0.76 | 1.65 | 0.93 | 0.62 | 1.39 |
| **Total household size (Reference: Less than 4)** | | | | | | |
| 4 or 5 | 0.73 | 0.53 | 0.99 | 0.95 | 0.69 | 1.32 |
| 6 or more | 0.72 | 0.51 | 1.00 | 0.90 | 0.63 | 1.29 |
| **Whether single parent (Reference: No)** | | | | | | |
| Yes | 1.66 | 1.18 | 2.33 | 1.28 | 0.89 | 1.85 |
| **Quality of relationship with partner (Reference: Excellent)** | | | | | | |
| Good | 1.93 | 1.45 | 2.57 | 1.82 | 1.35 | 2.44 |
| Average to poor | 3.64 | 2.44 | 5.43 | 5.07 | 3.37 | 7.62 |
| **Loneliness (Reference: Not lonely)** | | | | | | |
| Lonely most/all of the time | 8.37 | 5.70 | 12.27 | 8.50 | 5.71 | 12.65 |
| **Social support (Reference: Easy to get support)** | | | | | | |
| Not easy to get support | 2.25 | 1.78 | 2.86 | 2.13 | 1.67 | 2.73 |
| **Food security (Reference: Secure)** | | | | | | |
| Insecure—often true that food does not last | 3.33 | 2.09 | 5.28 | 3.46 | 2.15 | 5.58 |
| **Housing security (Reference: Secure)** | | | | | | |
| Insecure—worried about eviction/repossession | 3.29 | 2.36 | 4.58 | 3.00 | 2.11 | 4.25 |
| **Job security: whether expecting the main earner to still have a job in 12 months (Reference: Yes)** | | | | | | |
| No | 0.84 | 0.49 | 1.43 | 0.96 | 0.56 | 1.64 |
| **Housing condition (Reference: No damp in the house)** | | | | | | |
| Damp in the house | 1.85 | 1.45 | 2.38 | 1.57 | 1.21 | 2.03 |
| **Frequency of physical activity (Reference: Every day)** | | | | | | |
| Most days | 1.58 | 1.15 | 2.18 | 1.67 | 1.21 | 2.31 |

Continued

**Table 4** Continued

| Exposure | PHQ-8 increase ≥5 | | | GAD-7 increase ≥5 | | |
|---|---|---|---|---|---|---|
| | OR | Low 95% CI | High 95% CI | OR | Low 95% CI | High 95% CI |
| 1 or 2 days/week | 1.93 | 1.40 | 2.67 | 1.67 | 1.20 | 2.31 |
| Never | 3.13 | 2.15 | 4.56 | 2.55 | 1.72 | 3.78 |
| Change in frequency of physical activity since lockdown (Reference: More than before) | | | | | | |
| About the same | 0.66 | 0.45 | 0.97 | 0.82 | 0.56 | 1.21 |
| Less than before | 1.27 | 0.94 | 1.72 | 1.43 | 1.04 | 1.95 |
| Time between baseline and COVID-19 lockdown survey | | | | | | |
| Time (in months) | 1.00 | 0.99 | 1.01 | 1.00 | 0.99 | 1.01 |
| Cohort source (Reference: BiBGU) | | | | | | |
| BiBBS | 0.84 | 0.64 | 1.10 | 0.84 | 0.63 | 1.12 |

BiBBS, Born in Bradford's Better Start; BiBGU, Born in Bradford's Growing Up.

For Pakistani heritage mothers living in large households, the odds of an increase in depression was reduced (OR=0.54, 95% CI 0.31 to 0.94) compared with White British mothers for whom there was no association (OR=0.91, 95% CI 0.47 to 1.73). There were no clear associations between ethnicity and household size for anxiety (OR=0.61, 95% CI 0.34 to 1.07; OR=1.13, 95% CI 0.58 to 2.20).

### Mothers Lived Experience
Free text responses to the question 'What are your three biggest worries at the moment?' were available for 1799 mothers. Only a small proportion of women identified their mental health issues as one of their biggest worries, n=105 (6%, 95% CI 5% to 7%), slightly greater in White British mothers, n=51 (8%, 95% CI 6% to 11%) than in mothers of Pakistani heritage, n=32 (4%, 95% CI 3% to 5%). More often, mothers reported how wider issues and concerns impacted on their mental health and well-being.

### Health anxieties about COVID-19
The most commonly reported worry was fear of bringing the virus home (eg, from the shops or from their places of work), and themselves or members of their family becoming ill or dying, as well as the fear of what would happen to their children if this did happen to them.

> I worry about contracting coronavirus particularly whilst at work and either becoming critically myself unwell or bringing it home to my family.

> Feeling anxious about the virus and constantly worrying about my kids which 2 of them have health issues and are quite vulnerable.

> I feel particularly anxious to even step out of the house even for food shopping or taking a walk/exercise.

> I worry how this will affect my children. I'm terrified they will be separated as I have 2 children with my ex-husband and one with my current. So I haven't been outside in 10 weeks.

### Mental load
Mothers often reported the mental load of managing work, home schooling, childcare and domestic tasks, without the break provided by children attending school, nursery or other childcare. Being or feeling stuck inside and unable to move around freely contributed to a sense of suffocation and feeling overwhelmed, and many mothers acknowledged that this was having a detrimental effect on their mental health and self-esteem:

> I'm worried [about] having a nervous breakdown or a panic attack…can't get a break from all the responsibilities and go somewhere for fresh air even.

> Finding working from home and looking after children very demanding. No time alone. No silence. Surrounded by people and electronics all my waking hours.

**Table 5** ORs (with 95% CI) from unadjusted logistic regression models for an increase of 5 points or more on eight-item Patient Health Questionnaire (PHQ-8) between pre-COVID-19 and COVID-19 lockdown surveys for White British and Pakistani heritage mothers

| | White British | | | Pakistani heritage | | |
|---|---|---|---|---|---|---|
| Exposure | OR | Low 95% CI | High 95% CI | OR | Low 95% CI | High 95% CI |
| **Financial security (Reference: Living comfortably)** | | | | | | |
| Doing alright | 2.00 | 1.12 | 3.56 | 1.70 | 0.88 | 3.27 |
| Just about getting by | 3.98 | 2.15 | 7.38 | 3.03 | 1.56 | 5.88 |
| Finding it quite/very difficult | 12.14 | 5.15 | 28.60 | 5.79 | 2.90 | 11.56 |
| **Food security (Reference: Secure)** | | | | | | |
| Insecure | 3.73 | 1.58 | 8.77 | 4.13 | 2.08 | 8.17 |
| **Housing security (Reference: Secure)** | | | | | | |
| Insecure | 4.14 | 2.07 | 8.29 | 3.20 | 1.99 | 5.16 |
| **Loneliness (Reference: Not lonely)** | | | | | | |
| Lonely most/all of the time | 7.17 | 3.66 | 14.01 | 11.22 | 6.45 | 19.53 |
| **Social support (Reference: Easy to get support)** | | | | | | |
| Not easy | 2.08 | 1.39 | 3.10 | 2.20 | 1.54 | 3.14 |
| **Quality of relationship with partner (Reference: Excellent)** | | | | | | |
| Good | 1.68 | 1.02 | 2.77 | 1.76 | 1.15 | 2.68 |
| Average to very poor | 2.61 | 1.31 | 5.20 | 4.91 | 2.78 | 8.67 |
| **Total household size (Reference: Less than 4)** | | | | | | |
| 4 or 5 | 0.69 | 0.44 | 1.07 | 0.61 | 0.35 | 1.05 |
| 6 or more | 0.91 | 0.47 | 1.73 | 0.54 | 0.31 | 0.94 |
| **Frequency of physical activity (Reference: Every day)** | | | | | | |
| Never | 5.54 | 2.69 | 11.43 | 3.34 | 1.88 | 5.92 |
| 1 or 2 days/week | 1.97 | 1.15 | 3.37 | 2.05 | 1.22 | 3.47 |
| Most days a week | 1.33 | 0.81 | 2.17 | 2.08 | 1.21 | 3.58 |

I am a keyworker who works five days of week, only while my children are at school. I am becoming mentally drained/exhausted constantly being on duty, it is impacting my mental health.

Homeschooling I am not cut out for this and don't feel good enough.

### Loss of social support

The loss of social support caused by lockdown, especially for those who did not live with their partner or were single parents, was highlighted as causing loneliness and isolation for some.

My support network was my partner, who I cannot see due to lockdown.

Not able to see my boyfriend—feel isolated and alone once children are in bed with no adult face to face interaction.

The isolation and loneliness has been a challenge and having to take care of my child on my own full-time.

Being a single parent of a disabled child I rely on my social life/friends for my mental wellbeing and I worry about when it'll be safe to see/hug my immediate family who live locally.

### Financial and employment insecurity

Household finances and the stress of unemployment or job insecurity related to lockdown measures were also a major worry for women. Many families were in debt, reliant on credit cards and benefits, and in insecure employment before the lockdown measures began and were only just about getting by prior to the pandemic.

I have struggled financially during this time. I would like to not worry about money and bills and shopping and outgoings. I have enough worries being a full time carer.

Worried about the financial impact of covid 19. I am currently furloughed from work but I worry that the virus will have an impact on the business. My husband is self-employed and is not eligible to any funds.

### Being unable to switch off

Participants described being frightened of the news reports but unable to switch off, and were wondering when, if ever, things would become normal again:

Worry about my mental health, I know I'm struggling. Worry about the future and how this will affect the country financially. Worry about my children's education & what they've missed out on.

**Table 6** ORs (with 95% CI) from unadjusted logistic regression models for an increase of 5 points or more on Generalised Anxiety Disorder Assessment (GAD-7) between pre-COVID-19 and COVID-19 lockdown surveys for White British and Pakistani heritage mothers

| Exposure | White British | | | Pakistani heritage | | |
|---|---|---|---|---|---|---|
| | OR | Low 95% CI | High 95% CI | OR | Low 95% CI | High 95% CI |
| **Financial security (Reference: Living comfortably)** | | | | | | |
| Doing alright | 1.70 | 0.97 | 3.00 | 0.93 | 0.50 | 1.74 |
| Just about getting by | 2.82 | 1.53 | 5.21 | 2.55 | 1.39 | 4.68 |
| Finding it quite/very difficult | 7.69 | 3.43 | 17.26 | 4.35 | 2.27 | 8.30 |
| **Food security (Reference: Secure)** | | | | | | |
| Insecure | 3.59 | 1.47 | 8.7 | 4.96 | 2.49 | 9.87 |
| **Housing security (Reference: Secure)** | | | | | | |
| Insecure | 3.32 | 1.67 | 6.57 | 3.07 | 1.84 | 5.10 |
| **Loneliness (Reference: Not lonely)** | | | | | | |
| Lonely | 5.90 | 3.03 | 11.47 | 11.27 | 6.16 | 20.56 |
| **Social support (Reference: Easy to get support)** | | | | | | |
| Not easy | 2.33 | 1.55 | 3.51 | 2.09 | 1.45 | 3.01 |
| **Quality of relationship with partner (Reference: Excellent)** | | | | | | |
| Good | 1.54 | 0.93 | 2.55 | 1.88 | 1.22 | 2.89 |
| Average to very poor | 4.39 | 2.26 | 8.53 | 4.99 | 2.72 | 9.14 |
| **Total household size (Reference: Less than 4)** | | | | | | |
| 4 or 5 | 1.01 | 0.63 | 1.62 | 0.64 | 0.36 | 1.14 |
| 6 or more | 1.13 | 0.58 | 2.20 | 0.61 | 0.34 | 1.07 |
| **Frequency of physical activity (Reference: Every day)** | | | | | | |
| Never | 3.90 | 1.82 | 8.36 | 2.50 | 1.40 | 4.45 |
| 1 or 2 days/week | 1.52 | 0.88 | 2.63 | 1.61 | 0.97 | 2.70 |
| Most days a week | 1.45 | 0.90 | 2.34 | 1.86 | 1.09 | 3.18 |

All the bad news on the TV, and the death rate on the News. All the information on the news makes me panic more.

When will this end? Will it end?

### A loss of coping strategies

For those who had existing mental health issues before lockdown, the lockdown measures had often taken away their sources of support, their normal routines and methods of coping. In addition, some respondents reported being unable to access mental health services due to COVID-19 and lockdown measures:

Mental health—Exercise at the gym was a coping mechanism, now closed. Can do bits at home but nothing like at the gym. Cannot get any gym equipment—out of stock.

Mental Health—struggling to motivate myself and to keep to a routine. Self-destructive behaviour—drink drugs and binge eating. I am missing my work and colleagues.

Not getting my mental health support since the lockdown. My CPN (Community Psychiatric Nurse) not returning my calls. It has made me a lot worse. I try to talk to my husband so I am not keeping everything inside.

Mental health, I have had previous issues in the past and am struggling and don't feel like I can approach my GP at the minute as it isn't an emergency.

### Positive aspects of lockdown

Many participants reported positive aspects to changes enforced by the lockdown, commenting that they were getting to spend more quality time with their children, were enjoying a slower pace of life, a more relaxed routine and spending less time driving and commuting.

Life has become a lot more relaxed over the last 3 weeks, no manic mornings trying to get everybody out of the house, time with kids, doing stuff with kids I would normally say 'not now' to. Get to know kids more. More time outside, [doing] jobs in the house that need doing.

Spending more time with my husband and feeling appreciative of each other and having a relaxed day with the children instead of running to school to mosque and then all the extra clubs it's a more relaxed day.

Ramadan is the easiest it has ever been, we are free to make up our sleep and not push ourselves too much, had time to do nice things during Ramadan including having a more peaceful time not having to

do school runs, be stressed out, my husband had a chance to take a slower pace to life and not get too stressed.

## DISCUSSION

We compared depression and anxiety during the March–June 2020 UK lockdown to pre-COVID-19 depression and anxiety data collected in our longitudinal birth cohort studies. We found that clinically important symptoms of depression and anxiety increased from 11% to 20% and from 11% to 16%, respectively. These findings reflect those of other longitudinal studies which have reported similar changes in mental health from before to during the COVID-19 pandemic.[3–5]

We hypothesised that our vulnerable population with diverse ethnicity and high levels of deprivation would be susceptible to increases in depression and anxiety during the lockdown. Financial, food and housing insecurity all increased the odds of an increase in depression and anxiety, as did loneliness, a lack of social support, an average/poor partner relationship and a lack of physical activity.

There was no clear association between White British and Pakistani heritage mothers and a clinically important increase in depression or anxiety. However, when we separated out the regression analyses by ethnicity, we found interesting differences in the magnitude of the associations with an increase in depression and anxiety: mothers from a Pakistani heritage had greater odds of an increase in depression and anxiety if they were lonely or had an average/poor relationship (for depression, but not anxiety) than White British mothers. Pakistani heritage mothers had a much reduced odds of an increase in depression if they lived in a large household compared with White British mothers (for whom there was no association). In contrast, mothers of White British ethnicity had greater odds of an increase if they were financially insecure and/or physically inactive compared with Pakistani heritage mothers reporting the same exposures.

The free text responses supported these findings, highlighting a complex interplay of challenges contributing to poor mental health in mothers including: acute health anxieties; the mental load of managing multiple roles and responsibilities; the loss of social support and other coping strategies; pressures of financial and employment insecurity; and being unable to switch off from the pandemic.

The potential ethnic differences in the magnitude of the associations of different variables and increased mental ill health reported in this study warrant further investigation, including an understanding of potentially differing protective factors in different ethnic groups. We have previously reported, outside of the COVID-19 context, that (a) White British mothers are more likely to have their mental ill health identified by health professionals than South Asian mothers, and that (b) both White British and South Asian mothers are equally likely

to disclose symptoms in self-report research questionnaires, as used here.[23 24] Ethnic differences in household structure and cultural practices might provide more support in times of adversity, as we have found in the case of food insecurity previously.[25 26]

This is a longitudinal study comparing data collected before the COVID-19 pandemic and during the March–June 2020 lockdown which has allowed us to explore change over that time period. It also provides findings from a highly ethnically diverse population, the majority of whom live in the most deprived centiles in the UK. We are not aware of other studies that have explored longitudinal change in mental health from pre-COVID-19 to during the COVID-19 lockdown in a similar population.

Respondents were mothers of children aged 0–5 and/or 9–13 which may limit the wider generalisability, though our findings are broadly similar to those from a previous longitudinal study of two UK cohorts that included adult men and women (not all of whom were mothers), and found the increased risk of poor mental health in lockdown to be greater in women.[4] Our pre-COVID-19 measures were taken from data collected over the past 4 years, so we cannot, with confidence, attribute all changes to the pandemic and the lockdown. For example, it is possible that some of the difference reflects age-related change in the women and/or their children over time. It is also possible that we have underestimated some of the adverse impact of lockdown as a significant percentage of the BiBBS participants were pregnant at baseline, which itself is associated with raised levels of mental ill health. However, our analysis did not find increased levels of depression and anxiety for the BiBBS cohort compared with the BiBGU cohort. Similarly, our analyses did not find any association between time since baseline data capture and the odds of an increase in depression and anxiety, suggesting that the timing of our baseline collection is not influencing our findings.

We undertook unadjusted logistic regression analyses to explore possible factors that might explain or influence changes in mental health from before to during the pandemic, and repeated these separating out the two main ethnic groups. This approach avoids the difficulties inherent in interpreting the ethnicity coefficient in regression models controlling for other variables[20]; however, it also limits the interpretation of the data. In addition, the total sample size for this mental health analysis was ~1700 mothers and many estimates in the analyses were, as a consequence, imprecise with wide CIs. These results demonstrate the need for further studies with sufficient sample sizes, including boosted samples of ethnic minority groups to enable accurate understanding of the possible differing experiences and needs of the diverse UK population.

While it is possible that our results are influenced by the survey response rate of 28%, these participants were representative of the BiBGU and BiBBS cohorts,[15] including on the rates of baseline depression and

anxiety scores, and have demonstrated a wide variability in most characteristics (table 1).

We undertook stratified analyses to explore ethnic differences in how exposures related to changes in mental health from before to during the pandemic. However, while our analysis was able to look for differences between Pakistani heritage and White British mothers, we were unable to explore the heterogeneity of the 'other' ethnic group and we have no doubt missed important nuances in this population based on different social and cultural experiences.

We will continue to follow the BiBGU and BiBBS families so that we can look at trajectories of change over time, including during any future national or regional lockdowns. At the time of writing (November 2020), the UK was under a further nationwide lockdown, following a period of reduced restrictions during summer 2020 and tighter regional restrictions in the more deprived cities of Northern England, including Bradford, through the autumn. The impact of further and longer periods of restrictions and lockdowns will be a focus of our study moving forward. As the longitudinal follow-up lengthens and more data become available, it may be possible to model different transitions in mental health in relation to changing social and economic circumstances. It may also be of interest to explore positive/null transitions from clinically important symptoms to mild/no symptoms, or continued mild/no symptoms despite reported exposure to adversity.

Our results highlight the potential public health impact of lockdown on mental health, particularly in those who are lonely and economically insecure. Mental health problems are in general less visible than physical symptoms and in particular the physical symptoms related to COVID-19, including the acutely ill patients in Intensive Care Units, but may have more significant longer term consequences. This study also suggests a complex interplay of factors between an individual's circumstances and the odds of worsening mental health during the pandemic. A 'one-size-fits-all' approach to supporting mental ill health will not be effective; instead, understanding and addressing the potential causes in different groups will be important. For example, in case of future need for social distancing measures, government and local councils should consider policies that permit 'social bubbles' that can be implemented to reduce loneliness for those at risk of mental health problems, and they as well as voluntary services should continue to focus support to those who are lonely/isolated. Policy and decision makers should also make provision for the continuing need to support and protect vulnerable families from financial, food and housing insecurity, all of which were associated with poor mental health in this study. These actions will be important throughout any further regional and national lockdowns and during the postpandemic period of recovery.

**Author affiliations**
[1]Bradford Institute for Health Research, Bradford Teaching Hospitals NHS Foundation Trust, Bradford, UK
[2]Health Sciences, University of York, York, UK
[3]Hull York Medical School, University of York, York, UK
[4]Wolfson Institute for Population Health, Queen Mary University of London and Barts and The London School of Medicine and Dentistry, London, UK
[5]MRC Integrative Epidemiology Unit, The University of Bristol, Bristol, UK
[6]Population Health Science, University of Bristol Medical School, Bristol, UK
[7]Faculty of Life Sciences, University of Bradford, Bradford, UK

**Acknowledgements** Born in Bradford (BiB) is only possible because of the enthusiasm and commitment of the children and parents in BiB. We are grateful to all the participants, parent governors and community research advisory group members, schools, health professionals and researchers who have made BiB happen.

**Collaborators** Bradford Institute for Health Research COVID-19 Scientific Advisory Group: Dr Bo Hou, Dr Gillian Santorelli, Dr Jane West, Kuldeep Sohal, Dr Laura Sheard, Professor Mark Mon-Williams, Dr Michael McCooe, Dr Tom Lawton.

**Contributors** JD was involved in the concept and study design, design of the data collection tools, overall supervision of the study and data collection; wrote the statistical analysis plan; and drafted and revised the paper. She is the guarantor of the study. BK was involved in the design of the data collection tools, wrote the statistical analysis plan, conducted all statistical analyses, and drafted and revised the paper. BL devised the coding framework for free text responses, analysed the data, and drafted and revised the paper. SB was involved in the concept and study design, design of the data collection tools, supervision of the study and data collection, and reviewing and revising the paper. CC was involved in the concept and study design, and reviewing and revising the paper. KW was involved in design of the data collection tools, data curation, analysis, and reviewing and revising the paper. KS was involved in the study design, design of the data collection tools, supervision of the study and data collection, and reviewing and revising the paper. KC was involved in the study design, design of the data collection tools, supervision of the study and data collection, data curation, and reviewing and revising the paper. MB was involved in the study design, and drafted and revised the paper. NS advised on the clinical significance of changes in mental ill health, revising the statistical analysis plan, and drafted and revised the paper. TS was involved in the concept and study design, supervision and support to junior members of the team, and drafted and revised the paper. DL was involved in the concept and study design, design of the data collection tools, and drafted and revised the paper. JW was involved in the concept and study design, design of the data collection tools, supervision and support to junior members of the team, and drafted and revised the paper. RM was involved in the concept and study design, design of the data collection tools, revised the statistical analysis plan, and drafted and revised the paper. KP was involved in the concept and study design, design of the data collection tools, supervision and support to junior members of the team, revised the statistical analysis plan, and drafted and revised the paper.

**Funding** This study was funded by The Health Foundation COVID-19 Award (2301201), with further contributions from a Wellcome Trust infrastructure grant (WT101597MA); a joint grant from the UK Medical Research Council (MRC) and UK Economic and Social Science Research Council (ESRC) (MR/N024391/1); the National Institute for Health Research under its Applied Research Collaboration Yorkshire and Humber (NIHR200166); ActEarly UK Prevention Research Partnership Consortium (MR/S037527/1); Better Start Bradford through The National Lottery Community Fund; and the British Heart Foundation (CS/16/4/32482). DL works in a unit that received support from the University of Bristol and UK MRC (MC_UU_00011/6) and is a UK National Institute for Health Research senior investigator (NF-0616-10102).

**Disclaimer** The views expressed in this publication are those of the authors and not necessarily those of the National Institute for Health Research or the Department of Health and Social Care.

**Competing interests** All authors had financial support from funding bodies listed for the submitted work. DAL declares previous support from Roche Diagnostics and Medtronic for research unrelated to that presented here.

**Patient consent for publication** Not required.

**Ethics approval** This study involves human participants and was approved by the HRA and Bradford/Leeds Research Ethics Committee (substantial amendments to BiBGU 16/YH/0320 and BiBBS 15/YH/0455). Participants gave informed consent to participate in the study before taking part.

**Provenance and peer review** Not commissioned; externally peer reviewed.

**Data availability statement** Data are available upon reasonable request. Born in Bradford offers open access to their data resources. Available data and procedures to access this can be found at: https://borninbradford.nhs.uk/research/how-to-access-data/_

**Open access** This is an open access article distributed in accordance with the Creative Commons Attribution 4.0 Unported (CC BY 4.0) license, which permits others to copy, redistribute, remix, transform and build upon this work for any purpose, provided the original work is properly cited, a link to the licence is given, and indication of whether changes were made. See: https://creativecommons.org/licenses/by/4.0/.

**ORCID iDs**
Josie Dickerson http://orcid.org/0000-0003-0121-3406
Brian Kelly http://orcid.org/0000-0003-1834-2992
Katy Shire http://orcid.org/0000-0002-2093-181X
Maria Bryant http://orcid.org/0000-0001-7690-4098
Najma Siddiqi http://orcid.org/0000-0003-1794-2152
Trevor A Sheldon http://orcid.org/0000-0002-7479-5913
Deborah A Lawlor http://orcid.org/0000-0002-6793-2262
John Wright http://orcid.org/0000-0001-9572-7293
Rosemary RC McEachan http://orcid.org/0000-0003-1302-6675
Kate E Pickett http://orcid.org/0000-0002-8066-8507

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
