## [Reviewer comments · BMJ Open]

ARTICLE DETAILS

TITLE (PROVISIONAL)	"When will this end? Will it end?" The impact of the March-June 2020 UK Covid-19 lockdown response on mental health: a longitudinal survey of mothers in the Born in Bradford study.
AUTHORS	Dickerson, Josie; Kelly, Brian; Lockyer, Bridget; Bridges, Sally; Cartwright, Christopher; Willan, Kathryn; Shire, Katy; Crossley, Kirsty; Bryant, Maria; Siddiqi, Najma; Sheldon, T; Lawlor, Deborah; Wright, John; McEachan, Rosemary; Pickett, Kate

VERSION 1 – REVIEW

REVIEWER	Price, Anna Murdoch Childrens Research Institute, Policy and Equity Group
REVIEW RETURNED	03-Feb-2021

GENERAL COMMENTS	Thank you for the opportunity to review this paper describing the experience of mental health and its correlates during the first three months of the COVID-19 lockdown. Research that seeks to understand the experience of people living in adversity, and who are often excluded from research, is important. This is especially so in light of the COVID-19 pandemic, which appears to be extending and entrenching disadvantage in many countries. I have a few queries that relate to the analysis and interpretation: 1. In the description of the analysis, what does the following statement mean? "Each model controlled for one variable associated with such an increase in model 1". Table 1 shows no evidence of a relationship between variables such as the cohort, maternal age, number of children etc with the outcomes. However, the confidence intervals are not tight and it's plausible to think that these things could be related. In the manuscript, I couldn't tell if and how these variables were considered in the adjusted analyses and, if not, why not?2. Related to the above, time is an important factor in this study, and in several ways. For example, children are recruited in two age groups; women recruited in BiBBS completed the pre-COVID data during pregnancy; the range of time since most recent pre-COVID-19 data collection was different between cohorts and also ranged 1-52 months. Did the authors examine how length of time was related to the exposure/outcomes, and was it controlled for in the analyses?3. How do these analytic decisions relate to the interpretation of the findings? There is some discussion of the limitations in the Discussion, but I wonder if the consideration of pregnancy/age/time/SES could be considered in the analyses so the Discussion could address them more directly.
---

	4. (minor) Could Table 1 be simpler to read if the columns for the lockdown survey are positioned to the right of the pre-COVID-19 data? This way, I could compare across rows, and quickly see where the data were drawn from (e.g. pre/during)?
--	---

REVIEWER	Hannigan, A University of Limerick
-----------------	---------------------------------------

REVIEW RETURNED	08-Feb-2021
-------------

GENERAL COMMENTS	This paper reports on an important source of longitudinal data for ethnically diverse and more deprived populations. There are however a number of issues to address. It is mentioned in the discussion that the lockdown response rate is 28% and a reference provided to an already published paper, however it would be important to give the response rate in the abstract and also in the Methods. The rate is low and it is useful for readers to bear this in mind when evaluating the findings. A statement is made that these participants are representative of the original cohorts – this refers to Table 1 of reference 15? It would be helpful to clarify that this representativeness refers to age and ethnicity only. Given the focus on mental health in this paper, what is the rate of none, mild, moderate/severe depression and anxiety pre-Covid in those who responded during lockdown compared to those who didn't? Figure 1 shows us that overall the rate of moderate/severe depression has increased but there have been many different types of transitions for example almost half of those with moderate/severe depression pre Covid improved (to mild or none) during lockdown. It would be helpful to quantify the statement that some participants' levels of depression and anxiety have improved and a table of those transitions with percentages may be more useful than working it out from the numbers in the Figures. There seems to be inconsistency in the numbers reported in the Figures and the numbers reported in the text e.g. the number with moderate/severe depression pre-Covid is 203 in Figure 1 but it is given as 212 in the text? What is the n for calculating the rates in the text e.g. 212 (11%) to 349 (19%)? The outcome of interest in the models is clinically important increase in depression and anxiety (yes, no) but we are not given the number and rate of those with a clinically important increase in depression and anxiety. This is important information for the overall findings but also for the models. Many of the confidence intervals are wide and this uncertainty in estimates should be acknowledged. It would also be helpful to give a measure of goodness of fit of the final models presented. An additional descriptive table comparing the characteristics of those with and without a clinically important increase in depression and anxiety would be useful. It may be helpful to acknowledge other approaches to the statistical analysis of this dataset which would exploit the multiple types of transitions over time. Given the volume of free text comments, was any software used for coding? Does the section in the results represent all the themes that emerged? It may be helpful to give age and ethnicity of the mothers who provided the quotes used. Were the quotes for negative impacts specifically selected because of mental health emphasis? How many participants reported positive impacts?
--

	Some of the statements in the discussion could be more nuanced e.g. 'the deterioration in mental health is large'. Despite the significant vulnerabilities of this cohort, the rate of depression is similar to that reported nationally (is there an age and gender specific rate available nationally for comparison?) and the increase in anxiety is from 11 to 16% which given the vulnerabilities and the context, doesn't seem that large (the rate of depression doubled nationally?). There were also positive impacts and those whose mental health improved but without knowing the rate of those with a clinically important increase, these findings are difficult to put in context. Minor comments: The paper needs a final proof read – there are missing full stops, capital letters, use of multivariate instead of multivariable, use n for subsamples instead of N?
--	--

REVIEWER	Ford, Tamsin University of Cambridge, Psychiatry
REVIEW RETURNED	22-May-2021

GENERAL COMMENTS	Thank you for inviting me to review this analysis of change of time in two ethnically diverse cohorts - there is much to commend it, particularly as the authors state, the vast majority of data on mental health pertains to white populations. In particular the use of validated measures, the inclusion of qualitative data and that data was collected pre-pandemic are all strengths. Some revision is recommended before acceptance. 1/ Attention to table numbers is required - supplementary table 1 does not provide information about the ethnicity as stated in the text and I think is referring to Table 1 instead as supplementary Tables 1-9 are referred to accurately later in the text - all the other table numbers are incorrect as a result. 2/ There have been some recent indications from the national survey of children and young people's mental health and a large study in UK schools (recently published in Journal of the American Academy of Child and Adolescent Psychiatry), all collecting data in the last 5 years, that those of white British background have worse mental health than those from ethnic minorities - this is a change from earlier UK surveys but is worse noting - the statement in strengths and limitations could usefully be changed to indicate the mental health of people from ethnic minorities may differ from the White population, and has often been poorer. 3/ The Co-SPACE study of parents and the UKLS quote suggested that the parents of young children were particularly struggling - the age of the child (even if just by cohort) may be important and has not been explored. I would recommend that the authors analyse this variable also. 4/ The cohort is mainly Pakistani and White - with other groups collapsed out of necessity- it is a great deal more ethnically diverse than almost all studies I have met but the authors need to acknowledge more clearly that their analysis provides most useful data about how Pakistani mothers are coping compared to White mothers - subgroups within the other category may be having different experiences which might account for the lack of findings
--

	5/ the cohort is relatively small for an epidemiological study, which should be acknowledged and this will limit power - retention is good but a comparison of those from the cohort who had data and could be included compared to those who did not would be useful - Table 1 is large and could usefully be split into table S1 with the data that indicates who was and was not included on the baseline variables to support the statement that there was no difference and Table 1 in the main text which could present the post Lockdown data 6/ The statement that there was little missing data needs evidence to support it - either the lower and upper % missing across all variables or the n's with data in the table Otherwise this is an important and interesting paper and once these changes are instituted I would recommend it for publication.
--	---

VERSION 1 – AUTHOR RESPONSE

Reviewer: 1

Our responses to Reviewer 1's comments are as follows:

In the description of the analysis, what does the following statement mean? "Each model controlled for one variable associated with such an increase in model 1".

We have reviewed the objectives in the introduction (page 3) and the methods data analysis section (page 5) to clarify the purpose of our study and the methods we have used more clearly. This confusing statement has been removed.

Table 1 shows no evidence of a relationship between variables such as the cohort, maternal age, number of children etc with the outcomes. However, the confidence intervals are not tight and it's plausible to think that these things could be related. In the manuscript, I couldn't tell if and how these variables were considered in the adjusted analyses and, if not, why not?

The variables that we selected for inclusion in the unadjusted analysis were those that showed an association with an increase in mental ill health in our general findings paper. This did include Cohort, mothers' age, household size. Table 1 describes each of these variables and the sample characteristics of each one. Table 3 then shows the unadjusted Odds Ratios and CIs of all of these variables with an increase in mental health. The only variable that was controlled for was pre-Covid19 PHQ-8 and GAD-7.

As explained above, as we have such large ethnic diversity we then repeated this analysis separating out the two main ethnic groups (Pakistani heritage and White British) to look for differences in the magnitude of association of key variables and mental ill health.

Following BMJ's commitment to 'resign statistical significance' we have amended our results section (Page 6&7) and report on findings of potential public health relevance whilst acknowledging the large confidence intervals. We have also added to the limitations section in the discussion a note regarding the large confidence intervals (page 10).

3. Related to the above, time is an important factor in this study, and in several ways. For example, children are recruited in two age groups; women recruited in BiBBS completed the pre-COVID data during pregnancy; the range of time since most recent pre-COVID-19 data collection was different between cohorts and also ranged 1-52 months. Did the authors examine how length of time was related to the exposure/outcomes, and was it controlled for in the analyses?

In the unadjusted analysis we do use the variable “Cohort” which compares the BiBBS women who differ in 3 ways to the BiBGU data (a) baseline in pregnancy; b) longer time since baseline data capture; c) younger children). This exposure did demonstrate some association with depression and anxiety (a greater odds of an increase in BiBGU than BiBBS mothers). This has now been added to the results section (page 7), and the discussion (page 9).

Based on these comments, we have now also added ‘time since baseline data capture’ as a continuous variable into the unadjusted model in Table 3. The results show that this was not a variable associated with an increase in depression/anxiety.

How do these analytic decisions relate to the interpretation of the findings? There is some discussion of the limitations in the Discussion, but I wonder if the consideration of pregnancy/age/time/SES could be considered in the analyses so the Discussion could address them more directly.

Cohort (pregnancy @ baseline; age of child); Age; time of pre-pandemic data collection and Index of Multiple Deprivation are now all considered in the analysis, and in the discussion we note that there are limitations in the data that was available ‘pre-pandemic’, as well as limitations in the analytical methods that we have chosen (page 9).

5. (minor) Could Table 1 be simpler to read if the columns for the lockdown survey are positioned to the right of the pre-COVID-19 data? This way, I could compare across rows, and quickly see where the data were drawn from (e.g. pre/during)?

We have separated Table 1 into two Tables: Table 1 now describes the socio-demographics of participants and Table 2 describes the pre Covid-19 and Covid-19 variables.

Reviewer: 2

It is mentioned in the discussion that the lockdown response rate is 28% and a reference provided to an already published paper, however it would be important to give the response rate in the abstract and also in the Methods.

The rate is low and it is useful for readers to bear this in mind when evaluating the findings.

We have added the response rate for the original data sources into the results section (page 5).

2.2 A statement is made that these participants are representative of the original cohorts – this refers to Table 1 of reference 15? It would be helpful to clarify that this representativeness refers to age and ethnicity only. Given the focus on mental health in this paper, what is the rate of none, mild, moderate/severe depression and anxiety pre-Covid in those who responded during lockdown compared to those who didn't?

We have added in a new Supplementary table 1 that provides the comparison of those that did and did not complete the baseline questionnaire on ethnicity, age and baseline depression and anxiety scores. This confirms that the sample was also representative on these key variables, and this is now noted in the results section of the paper (page 5).

2.3 Figure 1 shows us that overall the rate of moderate/severe depression has increased but there have been many different types of transitions for example almost half of those with moderate/severe depression pre-Covid improved (to mild or none) during lockdown. It would be helpful to quantify the

statement that some participants' levels of depression and anxiety have improved and a table of those transitions with percentages may be more useful than working it out from the numbers in the Figures. We have quantified these statements in the text describing Figures 1&2 (page 6) and added in an additional Supplemental Table 2 that describes these data numerically as suggested.

2.4 There seems to be inconsistency in the numbers reported in the Figures and the numbers reported in the text e.g. the number with moderate/severe depression pre-Covid is 203 in Figure 1 but it is given as 212 in the text? What is the n for calculating the rates in the text e.g. 212 (11%) to 349 (19%)?

We have added into the text an explanation of the numbers reported in the figures and text: 1,760 participants had both pre-Covid19 and Covid-19 lockdown depression data and 1634 had pre-Covid19 and Covid-19 lockdown anxiety data (page 6)

2.5 The outcome of interest in the models is clinically important increase in depression and anxiety (yes, no) but we are not given the number and rate of those with a clinically important increase in depression and anxiety. This is important information for the overall findings but also for the models. We have added this data into the text on page 6: The number of participants who experienced an increase in depression / anxiety scores by 5 or more points, was 367 (21.2%) and 348 (21.3%) respectively.

Many of the confidence intervals are wide and this uncertainty in estimates should be acknowledged. This has been acknowledged in the results and added to the limitations section of the discussion (Page 10).

It would also be helpful to give a measure of goodness of fit of the final models presented. Please see the note to all Reviewers on page 1 - we have now removed the adjusted analysis so have not provided a goodness of fit measure here.

An additional descriptive table comparing the characteristics of those with and without a clinically important increase in depression and anxiety would be useful. We have not added this table as we are keen for readers to focus on the characteristics highlighted as associated with this increase in the regression models. The new analysis looking at the magnitude of association by ethnicity provides a more nuanced description of potential differences in characteristics of those with and without increased mental ill health (Table 4a & b).

It may be helpful to acknowledge other approaches to the statistical analysis of this dataset which would exploit the multiple types of transitions over time. We have added this to the discussion on page 10: Acknowledging the importance of considering different models of transitions such as transitions in mental health in relation to changing social and economic circumstances; and/or positive transitions from moderate/severe symptoms to mild/no symptoms.

Given the volume of free text comments, was any software used for coding?

A specific software was not used for coding, as the data were not overly complex and most responses were a single sentence in length. A clear coding structure was in place.

Does the section in the results represent all the themes that emerged?

Themes and quotes were selected specifically as being of relevance to this paper on mental health, the discussion of all themes that emerged are in our main findings paper. We have made this clearer in the methods section (page 4)

It may be helpful to give age and ethnicity of the mothers who provided the quotes used. We agree that this would be interesting, but we are unable to add ages / ethnicity to the quotes as this goes against what was approved in our protocol and ethics to protect confidentiality.

Were the quotes for negative impacts specifically selected because of mental health emphasis? How many participants reported positive impacts?

We asked participants to tell us about their 3 main worries, so the questions themselves encouraged negative responses. There were a huge number of worries expressed by our mothers, so the findings in this paper have not been selected with any bias. We did also ask about positive impacts of the lockdown, however, because our focus here is on a clinically important increase in symptoms we haven't explored these in depth here. They are however, discussed briefly on page 9, and in more depth in our previous main findings paper.

Some of the statements in the discussion could be more nuanced e.g. 'the deterioration in mental health is large'. Despite the significant vulnerabilities of this cohort, the rate of depression is similar to that reported nationally (is there an age and gender specific rate available nationally for comparison?) and the increase in anxiety is from 11 to 16% which given the vulnerabilities and the context, doesn't seem that large (the rate of depression doubled nationally?). There were also positive impacts and those whose mental health improved but without knowing the rate of those with a clinically important increase, these findings are difficult to put in context.

We have amended our discussion section based on these comments (see page 9-10), including a more reflective comparison of our findings to other longitudinal studies and more nuanced statements.

Minor comments: The paper needs a final proof read – there are missing full stops, capital letters, use of multivariate instead of multivariable, use n for subsamples instead of N?

Thank you, we have now given the paper a careful proof read and amended these errors throughout.

Reviewer: 3

1/ Attention to table numbers is required - supplementary table 1 does not provide information about the ethnicity as stated in the text and I think is referring to Table 1 instead as supplementary Tables 1-9 are referred to accurately later in the text - all the other table numbers are incorrect as a result. We have checked and amended accordingly. Thank you for noticing this

2/ There have been some recent indications from the national survey of children and young people's mental health and a large study in UK schools (recently published in Journal of the American Academy of Child and Adolescent Psychiatry), all collecting data in the last 5 years, that those of white British background have worse mental health than those from ethnic minorities - this is a change from earlier UK surveys but is worse noting - the statement in strengths and limitations could usefully be changed to indicate the mental health of people from ethnic minorities may differ from the White population, and has often been poorer.

Thank you for sharing these useful references. We have noted these findings and referenced the paper that looked before and during Covid-19 in the discussion (Page 9).

3/ The Co-SPACE study of parents and the UKLS quote suggested that the parents of young children were particularly struggling - the age of the child (even if just by cohort) may be important and has not been explored. I would recommend that the authors analyse this variable also.

In our unadjusted analysis we use the variable "Cohort" which compares the BiBBs and BiBGU participants. This is a proxy for the non-separable variables (baseline at pregnancy; having a pre-school / school aged child). We have clarified the meaning of this variable in the results section (page

4) and note that there was some association with a clinically important increase in depression or anxiety from before the pandemic. This was in the opposite direction to that expected – BiBBS mothers were less likely to have an increase in symptoms, but is also confounded by baseline during pregnancy. We have noted the limitations of our findings based on the pre-pandemic data available to us.

We also note that the Co-SPACE study has tracked changes in mental health during the pandemic and suggested that parents of children aged <10 were more stressed, but parents of older children were more depressed. As we follow-up our families we will be sure to continue to look for changes on these important variables and compare to this key study.

4/ The cohort is mainly Pakistani and White - with other groups collapsed out of necessity- it is a great deal more ethnically diverse than almost all studies I have met but the authors need to acknowledge more clearly that their analysis provides most useful data about how Pakistani mothers are coping compared to White mothers - subgroups within the other category may be having different experiences which might account for the lack of findings

Thank you this is a really important point, and has been amended in the methods and results sections of the paper, and the heterogeneity of the “other” group added to the study limitations (page 9).

5/ the cohort is relatively small for an epidemiological study, which should be acknowledged and this will limit power

We have added a note in the limitations section regarding this point.

- retention is good but a comparison of those from the cohort who had data and could be included compared to those who did not would be useful

We have added a comparison of the representativeness of respondents compared to the entire cohorts who were invited but did not respond in a new supplementary Table 1 which includes socio-demographic variables and pre-pandemic depression and anxiety (which were representative in our survey respondents).

- Table 1 is large and could usefully be split into table S1 with the data that indicates who was and was not included on the baseline variables to support the statement that there was no difference and Table 1 in the main text which could present the post Lockdown data

We have amended the tables as follows:

S1 is a comparison of those who did and did not respond.

Table 1 in the paper has now been split into two Tables: Table 1 describes the sample characteristics of participants and Table 2 describes the pre Covid-19 and Covid-19 changes of interest.

6/ The statement that there was little missing data needs evidence to support it - either the lower and upper % missing across all variables or the n's with data in the table

The numbers of data and % missing for all variables are located in Table 1.

Otherwise this is an important and interesting paper and once these changes are instituted I would recommend it for publication.

Thank you, we really appreciate your comments.

VERSION 2 – REVIEW

REVIEWER	Price, Anna Murdoch Childrens Research Institute, Policy and Equity Group
REVIEW RETURNED	27-Jul-2021

GENERAL COMMENTS	Thank you for the opportunity to re-review this manuscript. It presents a great deal of information, and offers a perspective of a group of families who are typically unrepresented in research. This is a valuable contribution in its own right and also as a reference for researchers and policy makers seeking to understand changes in mental health over the course of the Covid-19 pandemic, and how to identify and support those most in-need. I have some minor suggestions, which are mostly to do with the reporting of numbers:  - Abstract: It's not clear why the ORs and CIs are presented for loneliness and financial insecurity, but not for food & housing insecurity, lack of physical activity and poor relationship. Is it possible to add these in or clarify the language in the abstract to say why not? - Strengths and limitations points (p2): says age range of 0-5 but elsewhere it is 0-4. - Results: This section begins by describing the n=2144 who responded to the Covid-19 survey. It refers to the Supp Table 1, but Supp Table 1 describes the n=1860 who had complete surveys and linked data (so the proportions with Pakistani or White British heritage are slightly different to the text). Because there are several denominators used (e.g. in the 'Within mothers...' subsection, it refers to 1760 and 1634 for the mental health data), I wonder if it would be clearer for the reader to use the 1860 for the results Tables? If not, I found this section a bit hard to work through, so would love some clarification in the text to work from one denominator to the next. - Results: In the first para, it says that the baseline characteristics were broadly representative. The distribution by ethnicity appears different for those retained versus lost-to-follow-up. It also says the BiBBS and BiB cohorts are represented, but the breakdown by cohort is not in the Supp Table. Do you mean the combined cohort? - Results: The denominator/reference group for the proportions changes throughout the text. To flag this for the reader (which helps me when comparing between text and Tables), it would help to add the denominator to each number, e.g. for example 67% (n=759/XXX) who had no symptoms...; and 54% (n=109/XXXX) of those...367/XXX (21%) of mothers reported a.... and 348/XXX (21%) reported a clinically important increase... - Supp Table 1: Related to my points regarding consistency in numbers above, the 'completed' and 'not completed' columns exclude n=284 who are represented in the 'eligible' column. I gather that these are those who didn't have complete data within the 2144 respondents. I think they need to be in a column somewhere - either the complete or not complete or their own.
--

	- Supp Table 2: I think this is easier to interpret than the Figures 1 and 2 and that you could make it a main Table and drop the Figures if you wanted. - Discussion: in the sentence 'our results highlight the potential public health impact of lock down on mental health...' - do you need to specify White British? The nuance of the paper is one of its strengths - that there common and different vulnerabilities related to ethnicity. - The text uses a combination of Covid, Covid19 and Covid-19. Good luck!
--	---

REVIEWER	Hannigan, A University of Limerick
REVIEW RETURNED	27-Jul-2021

GENERAL COMMENTS	Thank you for the opportunity to re-review this manuscript. It presents a great deal of information, and offers a perspective of a group of families who are typically unrepresented in research. This is a valuable contribution in its own right and also as a reference for researchers and policy makers seeking to understand changes in mental health over the course of the Covid-19 pandemic, and how to identify and support those most in-need. I have some minor suggestions, which are mostly to do with the reporting of numbers:  - Abstract: It's not clear why the ORs and CIs are presented for loneliness and financial insecurity, but not for food & housing insecurity, lack of physical activity and poor relationship. Is it possible to add these in or clarify the language in the abstract to say why not? - Strengths and limitations points (p2): says age range of 0-5 but elsewhere it is 0-4. - Results: This section begins by describing the n=2144 who responded to the Covid-19 survey. It refers to the Supp Table 1, but Supp Table 1 describes the n=1860 who had complete surveys and linked data (so the proportions with Pakistani or White British heritage are slightly different to the text). Because there are several denominators used (e.g. in the 'Within mothers...' subsection, it refers to 1760 and 1634 for the mental health data), I wonder if it would be clearer for the reader to use the 1860 for the results Tables? If not, I found this section a bit hard to work through, so would love some clarification in the text to work from one denominator to the next. - Results: In the first para, it says that the baseline characteristics were broadly representative. The distribution by ethnicity appears different for those retained versus lost-to-follow-up. It also says the BiBBS and BiB cohorts are represented, but the breakdown by cohort is not in the Supp Table. Do you mean the combined cohort? - Results: The denominator/reference group for the proportions changes throughout the text. To flag this for the reader (which helps me when comparing between text and Tables), it would help
---

	to add the denominator to each number, e.g. for example 67% (n=759/XXX) who had no symptoms...; and 54% (n=109/XXXX) of those...367/XXX (21%) of mothers reported a.... and 348/XXX (21%) reported a clinically important increase... - Supp Table 1: Related to my points regarding consistency in numbers above, the 'completed' and 'not completed' columns exclude n=284 who are represented in the 'eligible' column. I gather that these are those who didn't have complete data within the 2144 respondents. I think they need to be in a column somewhere - either the complete or not complete or their own. - Supp Table 2: I think this is easier to interpret than the Figures 1 and 2 and that you could make it a main Table and drop the Figures if you wanted. - Discussion: in the sentence 'our results highlight the potential public health impact of lock down on mental health...' - do you need to specify White British? The nuance of the paper is one of its strengths - that there common and different vulnerabilities related to ethnicity. - The text uses a combination of Covid, Covid19 and Covid-19. Good luck!
--	---

REVIEWER	Ford, Tamsin University of Cambridge, Psychiatry
REVIEW RETURNED	25-Jul-2021

GENERAL COMMENTS	There is so little robust population based data from the UK, and particularly from people who are of ethnic minority backgrounds that this paper is really important. The authors have responded to nearly all of the comments raised by the three previous reviewers, which has strengthened the paper but also raises some new issues that need to be addressed before I would recommend that the paper is published Specically, the method is now much clearer, as are the presentation of the results. However, confidence intervals should be placed around all prevalence estimates, particularly when statements about their size are being made - having withdrawn the reference to statistically significance, there are now some estimates that we cannot judge that lack confidence intervals as listed below:-  - first line of results in the abstract -study sample prevalence change pdf page 7 -page 8 of the pdf, cohort difference in increased anxiety or depression Good to see more detail about the thematic analysis of the open text - given concerns about the wide confidence intervals on the comparison between ethnicities, the second paragraph of results in the abstract should be removed and these themes and the common predictors of poorer mental health listed instead My biggest concern is the over-interpretation of differences between mothers of Pakistani and White British heritage - the confidence intervals are broad and overlapping - where they cross
---

	unity, which they do on several comparisons then we cannot be sure that the difference is as appears from the point estimate eg - line 48 of page 6 of the pdf - the CIs for both anxiety and depression cross 1 so we cannot be sure that that either ethnicity is less or more likely to experience poorer mental health - this may have been statistically significant and probably relates to power but the data presented suggest no evidence of a difference for either condition The same error is repeated in relation to anxiety and large households on the top of page 7 page - 8 - there is so much overlap in the confidence intervals that the data do not support differences in experience between those of different ethnicities - furthermore the paragraph on large households that starts on line 36- the only confidence interval to exclude 1 relates to anxiety in women of pakistani origin - so we can assert that a large household seems to reduce their risks of anxiety but there is no evidence of difference for the other subgroups analysed. Given the lack of firm findings in the comparison of ethnic groups - it would make sense to make more of the predictors - actually there was more similarity than difference although a larger study (and this is a good argument for sufficient sample sizes and large ethnic boosts) might have been able to estimate such differences with more precision) similarly drawing in the qualitative data and how it relates to the findings about the predictors would make more sense and better reflect the findings Pierce et al, Lancet psychiatry 2020 - parents of young children emerged as a high risk group - also similar findings with financial insecurity - but without the ethnically diverse sample Minor comments Line 27 on page 8 - there is a typographical error in the confidence interval that needs checking line 45 page 8 - sentence needs "which" to join the phrases
--	---

VERSION 2 – AUTHOR RESPONSE

Reviewer: 1

Comments to the Author:

- Abstract: It's not clear why the ORs and CIs are presented for loneliness and financial insecurity, but not for food & housing insecurity, lack of physical activity and poor relationship. Is it possible to add these in or clarify the language in the abstract to say why not?

*These have now been added in to the abstract, see page 2.

- Strengths and limitations points (p2): says age range of 0-5 but elsewhere it is 0-4.

*0-5 is correct, and this has been amended throughout the paper.

- Results: This section begins by describing the n=2144 who responded to the Covid-19 survey. It refers to the Supp Table 1, but Supp Table 1 describes the n=1860 who had complete surveys and linked data (so the proportions with Pakistani or White British heritage are slightly different to the text). Because there are several denominators used (e.g. in the 'Within mothers...' subsection, it refers to 1760 and 1634 for the mental health data), I wonder if it would be clearer for the reader to use the 1860 for the results Tables? If not, I found this section a bit hard to work through, so would love some clarification in the text to work from one denominator to the next.

*We have clarified the first section of the results now (page 5), the descriptions of the sample were based on the 1860 in the study, but this was not clear in how we had presented this data. All data presented on demographics are based on the n=1860 used in this study / analysis.

- Results: In the first para, it says that the baseline characteristics were broadly representative. The distribution by ethnicity appears different for those retained versus lost-to-follow-up. It also says the BiBBS and BiB cohorts are represented, but the breakdown by cohort is not in the Supp Table. Do you mean the combined cohort?

*The reference to the BiBBS and BiB Cohorts has now been made clearer as it refers to the main results paper (Reference 15). The ethnic breakdown is skewed, and this has now been acknowledged in the results section (page 5).

- Results: The denominator/reference group for the proportions changes throughout the text. To flag this for the reader (which helps me when comparing between text and Tables), it would help to add the denominator to each number, e.g. for example 67% (n=759/XXX) who had no symptoms...; and 54% (n=109/XXXX) of those,....367/XXX (21%) of mothers reported a.... and 348/XXX (21%) reported a clinically important increase...

*We have added the denominator/reference group to the sub-headings of each section to help with interpretation of the findings. For the sample comparison it is n=1860; for the within-mother change it is n=1760 for depression and n=1634 Anxiety (page 6).

- Supp Table 1: Related to my points regarding consistency in numbers above, the 'completed' and 'not completed' columns exclude n=284 who are represented in the 'eligible' column. I gather that these are those who didn't have complete data within the 2144 respondents. I think they need to be in a column somewhere - either the complete or not complete or their own.

*We have added this additional information to Supplementary Table 1 now. Please note whilst we had age and ethnicity data for these 284, we did not have linked data so could not report on the depression or anxiety scores from baseline.

- Supp Table 2: I think this is easier to interpret than the Figures 1 and 2 and that you could make it a main Table and drop the Figures if you wanted.

*We have added this to the main paper as Table 3, and moved the figures to supplementary files.

- Discussion: in the sentence 'our results highlight the potential public health impact of lock down on mental health...' - do you need to specify White British? The nuance of the paper is one of its strengths - that there common and different vulnerabilities related to ethnicity.

* Based on the comments of reviewer 3, we have amended this statement so as not to over-interpret this finding. We note instead the potentially interesting changes in magnitude of associations between the 2 key ethnic groups (page 9).

- The text uses a combination of Covid, Covid19 and Covid-19.

*We have replaced all with Covid-19.

Reviewer: 3

- confidence intervals should be placed around all prevalence estimates, particularly when statements about their size are being made - having withdrawn the reference to statistical significance, there are now some estimates that we cannot judge that lack confidence intervals as listed below:-

*We have added in these CIs as requested to:

- first line of results in the abstract (see page 2)

-study sample prevalence change pdf page 7

-page 8 of the pdf, cohort difference in increased anxiety or depression

*This statement has now been removed from the results section at the request of the reviewer due to CIs overlapping 1 so CI addition is no longer relevant.

-Good to see more detail about the thematic analysis of the open text - given concerns about the wide confidence intervals on the comparison between ethnicities, the second paragraph of results in the abstract should be removed and these themes and the common predictors of poorer mental health listed instead

*This has now been done (see page 2).

-My biggest concern is the over-interpretation of differences between mothers of Pakistani and White British heritage - the confidence intervals are broad and overlapping - where they cross unity, which

they do on several comparisons then we cannot be sure that the difference is as appears from the point estimate

eg - line 48 of page 6 of the pdf - the CIs for both anxiety and depression cross 1 so we cannot be sure that either ethnicity is less or more likely to experience poorer mental health - this may have been statistically significant and probably relates to power but the data presented suggest no evidence of a difference for either condition

The same error is repeated in relation to anxiety and large households on the top of page 7

page - 8 - there is so much overlap in the confidence intervals that the data do not support differences in experience between those of different ethnicities - furthermore the paragraph on large households that starts on line 36- the only confidence interval to exclude 1 relates to anxiety in women of pakistani origin - so we can assert that a large household seems to reduce their risks of anxiety but there is no evidence of difference for the other subgroups analysed.

Given the lack of firm findings in the comparison of ethnic groups - it would make sense to make more of the predictors - actually there was more similarity than difference although a larger study (and this is a good argument for sufficient sample sizes and large ethnic boosts) might have been able to estimate such differences with more precision)

similarly drawing in the qualitative data and how it relates to the findings about the predictors would make more sense and better reflect the findings

*We have amended our findings and withdrawn any statements regarding association where the CIs cross 1 in the abstract, results and the discussion. As the findings section focuses on where associations were found, we have removed the statement relating to the cohort comparison (see page 7) as those CIs cross 1. We have left in the point regarding ethnicity as this was a key objective of our analysis but have changed the statement to say there was no clear association. We have also amended the statement regarding ethnicity and household size on page 7.

*In the discussion we raise your important note about the need for large sample sizes and ethnic boosts to research studies.

- Pierce et al, Lancet psychiatry 2020 - parents of young children emerged as a high risk group - also similar findings with financial insecurity - but without the ethnically diverse sample

*We have noted this, and corrected our introduction accordingly (page 3)

Minor comments

-Line 27 on page 8 - there is a typographical error in the confidence interval that needs checking

*This has been corrected.

-line 45 page 8 - sentence needs "which" to join the phrases

* We have proof read the page but cannot see where this additional connector needs to be placed.

Good luck!

*Thank you!

VERSION 3 – REVIEW

REVIEWER	Price, Anna Murdoch Childrens Research Institute, Policy and Equity Group
REVIEW RETURNED	13-Oct-2021

GENERAL COMMENTS	The paper is reading well, I have no further comments.
--

REVIEWER	Ford, Tamsin University of Cambridge, Psychiatry
REVIEW RETURNED	25-Oct-2021

GENERAL COMMENTS	Thank you for responding to the comments from the reviewers - this paper is now acceptable for publication
--